# The Effect of Zoledronic Acid on Serum Biomarkers among Patients with Chronic Low Back Pain and Modic Changes in Lumbar Magnetic Resonance Imaging

**DOI:** 10.3390/diagnostics9040212

**Published:** 2019-12-04

**Authors:** Katri Koivisto, Jaro Karppinen, Marianne Haapea, Jyri Järvinen, Eero Kyllönen, Osmo Tervonen, Jaakko Niinimäki, Mauro Alini, Jeffrey Lotz, Stefan Dudli, Dino Samartzis, Juha Risteli, Marja-Leena Majuri, Harri Alenius, Sibylle Grad

**Affiliations:** 1Medical Research Center Oulu, Oulu University Hospital and University of Oulu, 90220 Oulu, Finland; jaro.karppinen@ttl.fi (J.K.); marianne.haapea@oulu.fi (M.H.); jyri.jarvinen@ppshp.fi (J.J.); eero.kyllonen@oulu.fi (E.K.); marjaleena.majuri@gmail.com (M.-L.M.); 2Center for Life Course Health Research, University of Oulu, 90220 Oulu, Finland; 3Finnish Institute of Occupational Health, 90220 Oulu, Finland; 4Research Unit of Medical Imaging, Physics and Technology, University of Oulu, 90220 Oulu, Finland; osmo.tervonen@oulu.fi (O.T.); jaakko.niinimaki@oulu.fi (J.N.); 5AO Research Institute Davos, 7270 Davos Platz, Switzerland; mauro.alini@aofoundation.org (M.A.); sibylle.grad@aofoundation.org (S.G.); 6Department of Orthopaedic Surgery, University of California San Francisco, San Francisco, CA 94143, USA; jeffrey.lotz@ucsf.edu; 7Center of Experimental Rheumatology, University Hospital Zurich, 8008 Zurich, Switzerland; stefan.dudli@usz.ch; 8Department of Orthopedic Surgery, RUSH University, Chicago, IL 60612, USA; Dino_Samartzis@rush.edu; 9Northern Finland Laboratory Centre NordLab, Oulu University Hospital, 90220 Oulu, Finland; juha.risteli@oulu.fi; 10Institute of Environmental Medicine, Karolinska Institute, 17177 Stockholm, Sweden; harri.alenius@ki.se

**Keywords:** serum biomarkers, magnetic resonance imaging, Modic change, randomized trial, zoledronic acid, chronic low back pain

## Abstract

The aim of the current study was to compare changes in serum biomarkers, including inflammatory mediators, signaling molecules, growth factors and markers of bone turnover after a single intravenous infusion of 5 mg zoledronic acid (ZA, a long-acting bisphosphonate; *n* = 20) or placebo (*n* = 20) among patients with Modic changes (MC) and chronic low back pain in a randomized controlled design. The MCs were classified into M1, predominating M1, predominating M2, and M2. We measured the serum concentrations of 39 biomarkers at baseline, and one month and one year after treatment. After Benjamini–Hochberg (B–H) correction, we observed significant differences in three biomarkers over one year: Interferon-γ-inducible protein (IP-10) had risen in the ZA group (*p* = 0.005), whereas alkaline phosphatase (AFOS) and intact procollagen I N-terminal propeptide (iPINP) had significantly decreased in the ZA group, but had not changed in the placebo group (*p* < 0.001 for both). Change in iPINP correlated with change in the volume of all MC and M1 lesions. ZA downregulated bone turnover markers as expected and, surprisingly, increased the chemokine IP-10 relative to placebo treatment. This adds to our knowledge of the effects of ZA on MC and the biomarkers that signal this process.

## 1. Introduction

A Modic change (MC) is a vertebral bone marrow change visible in magnetic resonance imaging (MRI). Three different types of MC have been described, of which Type 1 (M1) shows fibrovascular replacement of the bone marrow and represents an inflammatory lesion; Type 2 MC (M2) shows fatty replacement of the red bone marrow; and Type 3 MC (M3) demonstrates subchondral bone sclerosis [1,2,3,4]. MC is associated with chronic low back pain (CLBP) [5,6,7] and M1 shows a stronger association with CLBP than other MC types [8,9,10]. In a Finnish patient study, conversion from M1 to M2 over two years was associated with improvement of pain symptoms [11]. Similar findings have been obtained among Danish patients [12].

Both infectious or autoimmune etiologies have been suggested as causes of MC, and both presuppose structural damage of the endplate [13,14,15]. However, improved diagnostic tools are needed to detect early MC lesions. One novel pre-clinical technique that holds great clinical potential is nuclear magnetic resonance spectroscopy, which is a noninvasive method to quantify Cutibacterium (formerly Propionibacterium) acnes load on the intervertebral discs [16,17]. There is also an urgent need to develop other MC biomarkers in order to increase early diagnosis and the choice of the right treatment and establishment of MC prognosis [13].

A biomarker is defined as a characteristic that is objectively measured and evaluated as an indicator of normal biologic or pathogenic processes, or pharmacologic responses to a therapeutic intervention [18]. Molecular diagnostic techniques have developed in recent years and may be helpful in revealing diseases at their early stage. However, only two studies of MC serum biomarkers have been published so far. In the first, serum high-sensitive C-reactive protein (hs-CRP) was higher among patients with M1 than among those with M2, or those without MC but with CLBP [19]. Another small French case-control study in turn found no difference in bone mineral density and bone remodeling marker levels (serum C-terminal crosslinked type 1 telopeptide, procollagen type 1 N-terminal propeptide, osteocalcin, and bone-specific alkaline phosphatase) among chronic LBP patients with M1 or without MC [20]. We have previously demonstrated how a single intravenous infusion of 5 mg zoledronic acid (ZA) decreased the intensity of pain among patients with CLBP in the short-term in comparison to placebo [21]. We hypothesized that the beneficial effect of ZA on symptoms may be partly due to the reduction of inflammation, which in turn may be reflected by serum biomarkers related to inflammation and bone turnover. Therefore, the objective of our study was to compare the effect of a single intravenous infusion of 5 mg ZA to that of placebo infusion on the change in the serum biomarker profile of patients with MC, and to evaluate whether serum biomarkers correlate with the type and size of MC at baseline.

## 2. Materials and Methods

### 2.1. Study Design and Selection of Patients

The study population consisted of patients with CLBP and MC in lumbar MRI [21]. Inclusion criteria were LBP for at least three months, LBP intensity of at least six on the 10-cm Visual Analog Scale (VAS) or an Oswestry Disability Index (ODI) [22] of at least 30% and an MC (M1, M2, or M1/2) in MRI [21].

The Oulu University Hospital ethics committee approved the study protocol (121/2008, 18 August 2008). Informed consent was obtained from all participants (ClinicalTrials.gov NCT01330238).

### 2.2. Treatment Intervention

Patients were randomized into a ZA group with an infusion of 5 mg of ZA (*n* = 20) or a placebo group (saline infusion, *n* = 20). Before the administration of the infusion, all the patients received 600 mg of ibuprofen or 1 g of paracetamol orally to prevent potential acute phase reactions, and 100,000 units of Vitamin D (VigantolR) orally to avoid hypocalcemia [21].

### 2.3. Magnetic Resonance Imaging

The average interval between baseline imaging and infusion was four months (range 0.4 to 11.5 months). Follow-up imaging was performed on average 11.9 months (range 11 to 13 months) after infusion [23]. The MRI equipment of the baseline imaging included five 1.5 T units (GE Signa Twinspeed, General Electric Medical Systems, Milwaukee, WI, USA; Philips Achieva and Philips Intera, Philips Medical Systems, Eindhoven, The Netherlands; Siemens Avanto and Siemens Espree, Siemens Medical, Erlangen, Germany), a 0.34 T unit (Siemens Magnetom C, Siemens Medical, Erlangen, Germany) and a 0.23 T unit (Philips Panorama, Philips Medical Systems, Eindhoven, The Netherlands). The imaging sequences were sagittal T1-weighted (T1 W) turbo spin-echo (TSE) or fast spin-echo (FSE) with fluid attenuation inversion recovery (FLAIR), sagittal T2-weighted (T2 W) TSE/FSE, and short tau inversion recovery sequences (STIR). The specific imaging parameters have been described previously [23]. The MRI equipment of the one-year follow-up imaging included two 1.5 T units (GE Signa Twinspeed and GE Optima, General Electric Medical Systems, Milwaukee, WI, USA) and a 3 T unit (Siemens Skyra, Siemens Medical, Erlangen, Germany). Imaging protocols were established for clinical spine imaging.

### 2.4. Image Analysis

We analyzed the MRIs for type and volume of each MC from sagittal images at a clinical workstation (Neaview Radiology, version 2.23, Neagen Corporation, Oulu, Finland) [23]. MC type was assessed using T1 W and T2 W images as previously described [10]. Four MC type groups were created: M1 (100%), predominating M1 (M1/2 (65:35%)), predominating M2 (M1/2 (35:65%)) and M2 (100%). The first two groups were considered M1-dominant, and the latter two M2-dominant. M1 was defined as consisting of purely oedemic signal changes and M2 consisting of purely fatty signal changes. Predominating M1 was defined as mixed Type 1/2 MC with more oedemic signal changes and predominating M2 as Type 1/2 MC with more fatty signal changes. The MC area (cm^2^) was determined from T2 W images. The volume (cm^3^) of each MC lesion was calculated by multiplying the area with the spacing.

### 2.5. Analysis of Serum Biomarkers

Whole blood was drawn at noon from fasting patients using serum collection tubes prior to the intervention (baseline), at one-month and at one-year follow-up. We told all the participants not to exercise, eat, or drink the previous evening or the morning before the blood were to be taken. The blood samples were allowed to clot at room temperature for 30–60 min and then centrifuged at 2500× *g* for 10 min at 4 °C. The resulting serum supernatant was stored in aliquots, first at −20 °C, and were then transferred within 2–20 h to −70 °C. We took serum samples from 40 patients at three timepoints (*n* = 120) to assess the selected inflammatory and signaling molecules, growth factors and markers of bone turnover. We measured the serum concentrations of 50 biomarkers, and quantified 38 biomarkers using electro-chemiluminescent based multi-array immunoassays from MesoScale Discovery (MSD, Rockville, MD, USA), following the protocol of the manufacturer: Chemokine panel: Eotaxin, eotaxin-3, interferon-gamma-inducible protein (IP)-10, macrophage inflammatory protein (MIP)-1-alpha, MIP-1-beta, monocyte chemotactic protein (MCP)-1, MCP-4, macrophage-derived chemokine (MDC)-1, regulated on activation normal T-expressed and secreted (RANTES), thymus and activation regulated chemokine (TARC); Cytokine panel: Granulocyte–macrophage colony stimulating factor (GM-CSF), interleukin (IL)-5, interleukin (IL)-7, IL-12/23p40, IL-15, IL-16, IL-17A, tumor necrosis factor (TNF)-beta; Inflammation panel: Interferon (IFN)-gamma, IL-1-alpha, IL-1-beta, IL-2, IL-6, IL-8, IL-12p70, TNF-alpha, IL-4, IL-10, IL-13; Angiogenesis panel: Vascular endothelial growth factor (VEGF)-A, VEGF-C, VEGF-D, TEK receptor tyrosine kinase (Tie-2), fms related tyrosine kinase 1 (Flt-1), basic fibroblast growth factor (bFGF); and Vascular injury panel: Intercellular adhesion molecule (ICAM)-1, vascular cell adhesion molecule (VCAM)-1, serum amyloid A (SAA). To analyze the interleukin 1 receptor antagonist (IL-1RA), we used epidermal growth factor (EGF), hepatocyte growth factor (HGF), interferon-alpha (IFN-A), interleukin-2 receptor (IL-2R) and monokine induced by gamma-interferon (MIG-1), a Cytokine Human Magnetic Multi-Plex Panel (Thermo Fisher Scientific, Waltham, MA, USA), according to the manufacturer’s instructions. The receptor activator of NF-κB ligand (RANKL) was measured using a commercial Luminex single plex kit (Human RANKL single plex kit, MilliplexTM Map, Merck Millipore, Billerica, MA, USA), and soluble interleukin-1 receptor, type II (IL-1sRII) with a human IL-1RII Quantikine Elisa kit (R&D Systems, Minneapolis, MN, USA). All Luminex assays were performed using Luminex xMAP Technology (Bio-Plex 200 System; BioRad, Hercules, CA, USA). For the IL-1sRII Elisa assay, absorption was read at 450 nm using an ELISA plate absorbance reader, and the correction was made at 570 nm (Multiskan MS, Thermo Scientific, Vantaa, Finland). Plasma alkaline phosphatase activity (AFOS) was analyzed using the IFCC recommended enzymatic method (Advia, Siemens, Germany). Serum highly sensitive CRP (hs-CRP) was analyzed using a nephelometric instrument (BN ProSpec, Siemens, Germany). Intact procollagen I N-terminal propeptide (iPINP) and C telopeptide of type I collagen (CTX-I) analyses were performed using the IDS-iSYS Multi-Discipline Automated System (IDS, Bolton, UK). As the results of 11 biomarkers (EGF, GM-CSF, IL-1A, IL-1B, IL-1RA, IL-2, IL-4, IL-5, IL-10, IL-12p70, IL-13) were below the limit of quantification for more than half of the samples in the whole study, we excluded them from the final results.

### 2.6. Statistical Analyses

The background variables are presented as frequencies with proportions and means with standard deviations. We compared these variables of the ZA and placebo groups using the chi-square test or independent samples *t*-test, respectively. The serum biomarkers at baseline, one month and one year are separately presented as medians with interquartile range for the ZA and placebo groups. The change in serum biomarkers from baseline to one month and one year in the ZA and placebo groups, and in the M1- and M2-dominant MC were compared using Mann–Whitney’s U-test. The change in the serum biomarkers from baseline to one month and one year in the ZA and placebo groups, and in the M1- and M2-dominant MC were compared separately, using the Wilcoxon signed-rank test. We analyzed the correlation of the change in serum biomarkers from baseline to one year with a change in primary MC total volume, M1 volume and M2 volume, and with a change in LBP using Spearman’s rank correlation coefficient. The Benjamini–Hochberg (B–H) procedure, with a 10% false discovery rate, was used to correct for multiple comparisons. We used IBM SPSS Statistics, version 24.0 to conduct the analyses.

## 3. Results

### 3.1. Study Population

All 40 enrolled, eligible patients participated in the one-year follow-up. There were no statistically significant differences between the baseline demographic characteristics of the ZA and placebo groups (Table 1). The patients’ mean age was 50 (SD 8.3) and mean body mass index (BMI) 26.8 (SD 3.2). The median duration of LBP was 330 (IQR 216-365) days and the mean VAS score for LBP was 6.7 (SD 1.5). The baseline serum biomarker concentrations of the placebo and ZA groups did not differ significantly.

The results of 11 biomarkers were below the limit of quantification for more than half of the samples in the whole study and were thus excluded from the final results. Table 2 presents the results of all the 39 serum biomarkers analyzed. The significant changes are in bold in Table 2 and the markers that remained significant after B–H correction are marked with an asterisk. At one-month follow-up, IL-8 in the placebo group and intact procollagen I N-terminal propeptide (iPINP) in the ZA group had decreased, whereas IP-10, MCP-1, SAA, and hs-CRP had increased in the ZA group (Table 2, Appendix A). Compared to placebo, the decrease in biomarker concentration from baseline to one month was greater in the ZA group in terms of iPINP (median of change −9.3 vs. −0.7 ng/mL, *p* = 0.018) and in CTX-1 (−0.27 vs. −0.02 pg/mL, *p* = 0.009). At one month, there were no significant differences between the ZA and placebo groups after B–H correction.

At one-year follow-up, IL-16 in the placebo group, and Tie-2, AFOS and iPINP in the ZA group had decreased, whereas IP-10 and IFN-A had increased in the ZA group. During the one-year follow-up, one cytokine biomarker (IL-16) had decreased in the placebo group and two biomarkers (IP-10 and IFN-A) had increased in the ZA group. The change in IP-10 (decrease of –13.5 in the placebo group vs. increase of 61.1 pg/mL in the ZA group, *p* = 0.005) was significant after B–H correction. Two serum biomarkers (AFOS and iPINP) had not changed in the placebo group but had decreased in the ZA group (2.5 vs. −9.0 U/l, *p* < 0.001 and 1.0 vs. −19.1 ng/mL, *p* < 0.001; respectively). At one year, two biomarkers had significantly decreased in the ZA group after B–H correction: AFOS (71 to 57, *p* < 0.001) and iPINP (36 to 15, *p* < 0.001; Table 2).

Among the patients with M2 at baseline, TARC had increased significantly during the one-year follow-up, whereas among the patients with M1 at baseline, AFOS and iPINP had decreased significantly (Appendix A). In the ZA group, the changes in total volume of MC and M1 volume during the one-year follow-up correlated positively with the changes in iPINP (Spearman’s correlations (rho) 0.65 and 0.49). None of the other changes were significant and are presented in Appendix A. The changes in LBP intensity did not correlate statistically significantly with the changes in the concentration of any biomarker, as shown in Appendix A.

## 4. Discussion

In the present study, a single intravenous infusion of 5 mg ZA gave rise to different trends in the ZA and placebo groups, leading to significant differences at one-year follow-up in one chemokine (IP-10), which had increased in the ZA group and decreased in the placebo group, whereas two bone metabolism biomarkers (AFOS and iPINP) had decreased in the ZA group. Change in iPINP correlated with the change in the volume of all MC and M1 lesions.

We measured serum levels of AFOS, iPINP, and CTX-1 to evaluate bone formation and resorption. Of the bone biomarkers, iPINP decreased significantly at both one month and one year and AFOS at one year in the ZA group. The decreased concentrations in the bone biomarkers were not unexpected, as ZA has shown to suppress bone remodeling among post-menopausal women with osteoporosis [24,25]. The former [24] documented a 58% reduction in iPINP at one year after ZA administration among postmenopausal women with osteoporosis, whereas in the present study, iPINP concentration decreased by 19.4% at one month and 58.3% at one year, compared with 2.8% and 7.9% increases in the placebo group, respectively. A small French case-control study found no difference in bone mineral density and bone remodeling marker levels (serum C-terminal crosslinked type 1 telopeptide, procollagen type 1 N-terminal propeptide, osteocalcin and bone-specific alkaline phosphatase) among chronic LBP patients with M1 or without MC [20].

In the chemokine panel, IP-10 was elevated at one month and at one year, while MCP-1 was elevated at one month in the ZA group. Both IP-10 and MCP-1 have been linked to pain [26,27,28,29]. In animal models, ZA reduced the expression of a number of pro-inflammatory and angiogenic mediators, including MCP-1 [30], whereas in our study, MCP-1 was elevated in the ZA group.

In the pro-inflammatory panel, IFN-A was elevated after one year in the ZA group—although not significantly so after B–H correction. Bisphosphonates do not only inhibit osteoclasts; they have also shown to suppress the secretion of proinflammatory cytokines such as IL-1, TNF-A, and IL-6 [31]. Several studies have found upregulation of pro-inflammatory cytokines in endplates with MC [32,33,34] and local anti-inflammatory treatment with glucocorticoid has shown to have a short-term alleviating effect on LBP [35]. In our RCT, the improvement in LBP in the ZA group [21] could have been due to the general ability of bisphosphonates to regulate bone turnover by suppressing osteoclast activity [36] or to direct anti-inflammatory effects.

In the vascular injury panel, hs-CRP and SAA were elevated after one month in the ZA group (but not significantly after B–H correction), but at one year there was no difference. Some studies have shown that administration of bisphosphonates elevates the levels of circulating CRP, IL-6, TNF-A, and cortisol levels [37,38]. ZA treatment often causes an acute phase reaction with fever and flu-like symptoms [39]. The published studies of inflammation parameters in MC are based on tissue samples [32,40,41], but only a few have assessed serum biomarkers. Rannou et al. found that high-sensitivity C-reactive protein (hsCRP) levels were elevated among patients with M1 and CLBP [19]. An Australian study demonstrated that elevation of CRP and IL-6 during the acute phase of LBP was associated with better recovery, whereas individuals with elevated TNF had poorer prognoses [42]. It might be that the “window of opportunity” to detect inflammatory serum biomarkers only exists in the early phase of MC development and shortly after the infusion of bisphosphonates.

A recent Dutch systematic review found moderate evidence of a positive association between the pro-inflammatory biomarkers CRP and IL-6 and the severity of nonspecific LBP, and a positive association between TNF-A and the presence of nonspecific LBP [43]. If it were possible to determine a fingerprint in the serum of patients with LBP that is associated with MC, serum profiling could be used to assess the prognosis and choice of treatment. Khan et al. [44] emphasized the potential of diagnostic biomarkers to guide an era of individualized spine medicine for personalized therapeutics in the treatment of LBP. In our study, however, the changes in LBP intensity did not correlate with the changes in any biomarker’s concentration.

Because of their potent antiresorptive activity, bisphosphonates, including ZA, represent standards of care for the treatment of osteoporosis. Studies in vitro and in vivo have shown that, for example, pamidronate and clodronate inhibit the synthesis of proinflammatory cytokines [45,46]. It has been proposed that the development of MC is dependent on the structural disruption of the intervertebral disc or endplate, and the inflammatory potential of the disc and the capacity of the bone marrow to respond to higher inflammatory stimuli [13]. The source of pain is probably the chemical and mechanical stimulation of the nociceptors adjacent to the damaged endplates. It has also been suggested that a low virulent infection of anaerobic bacteria may induce edema and inflammation [47]. When we exclude the theory of anaerobic infection from the pathophysiology of MC, we notice that ZA has several potential mechanisms for acting in MC lesions and is a promising, well-known and well-tolerated drug for treating LBP associated with MC. In the present study we could not prove the downregulation of serum biomarkers, including cytokines and growth factors, after ZA administration among LBP patients with MC. This might be due to our small sample size.

The strength of the present study is its randomized study design. Further strengths are its complete follow-up with no dropouts and 100% adherence, as medication was administered intravenously. All the analyses were performed using established, validated measurement assays in high-quality laboratories. Longitudinal data enable a better view of the possible inflammatory component in LBP with MC. However, we also acknowledge some limitations in our study. The small sample size of this pilot study might be inadequate to demonstrate significant changes in biomarkers, and multiple testing might increase the rate of false positive results. Thus, our results should be interpreted with caution. The possible presence of comorbidities and adverse lifestyles may affect the proinflammatory mediators observed in the systemic circulation of patients. For example, obesity is associated with increased serum levels of IL-6 and TNF-A [48], and smoking is characterized by elevated serum levels of, for example, TNF-A and IL-1, as well as other cytokines [49].

## 5. Conclusions

The present data demonstrated that a single intravenous infusion of 5 mg ZA gave rise to different trends in serum biomarkers in the ZA and placebo groups, leading to significantly different regulations at one-year follow-up for one chemokine (IP-10), whereas two bone metabolism biomarkers (AFOS and iPINP) decreased in the ZA group. ZA treatment had an expected downregulating effect on bone turnover markers. This study found an association, a positive correlation, between a change in iPINP concentration and the change in total volume of MC and M1 volume. The changes in LBP intensity did not correlate with the changes in any biomarker’s concentration.

This is the first RCT to investigate ZA in MC and to observe how it is reflected in serum biomarkers. The evaluation of these biomarkers expands our understanding of the mechanism of MC and LBP and may thereby guide us to new diagnostic and therapeutic methods. It adds to our knowledge of the effects of ZA on MC and on the biomarkers signaling this process. We conclude that serum biomarkers can be used to monitor the treatment effect of ZA among LBP patients with MC. Larger studies are required to demonstrate the findings of this pilot study.

## Figures and Tables

**Table 1 diagnostics-09-00212-t001:** Characteristics of patients treated with zoledronic acid (ZA) and placebo. Group differences between age, body mass index (BMI), and Modic changes (MC) volumes were analyzed using the *t*-test. Otherwise, the chi-square test was used.

Variable	ZA	Placebo	*p*-Value
Age (years), mean (SD)	49.2 (9.3)	51.5 (7.3)	0.400
BMI (kg/m^2^), mean (SD)	26.1 (3.3)	27.4 (3.2)	0.203
Males, *n* (%)	15 (75.0%)	11 (55.0%)	0.320
Regular smoker, *n* (%)	5 (25.0%)	6 (30.0%)	>0.999
**Primary MC type,** ***n*** **(%)**			0.103
MI	3 (15.0%)	3 (15.0%)	
MI/II-1	14 (70.0%)	7 (35.0%)	
MI/II-2	2 (10.0%)	8 (40.0%)	
MII	1 (5.0%)	2 (10.0%)	
**Number of MCs, *n* (%)**			0.333
1	10 (50.0%)	14 (70.0%)	
2 or more	10 (50%)	6 (30.0%)	
**Volume of primary MC, mean (SD)**			
Type 1	7443 (4474)	5044 (3551)	0.068
Type 2	4542 (3897)	5868 (4840)	0.346
Total	11,985 (5140)	10,911 (5963)	0.546
Osteoarthritis, *n* (%)	1 (5.0%)	6 (30.0%)	0.091
Back pain (VAS), mean (SD)	6.6 (1.4)	6.8 (1.6)	0.646
Leg pain (VAS), mean (SD)	3.0 (3.1)	2.9 (2.3)	0.864
Oswestry disability index, mean (SD)	30.1 (11.0)	34.9 (9.8)	0.157

**Table 2 diagnostics-09-00212-t002:** Median concentrations and interquartile range (IQR) of serum biomarkers at baseline, one month and one year according to intervention (zoledronic acid (ZA) or placebo infusion). P^1^ shows the significance of the change in each respective biomarker concentration from baseline to one month or from baseline to one year. P^2^ shows the significance of the difference in change from baseline to one month or from baseline to one year in the concentration of each biomarker between the ZA and placebo groups. Significant *p*-values are in bold.

Serum Biomarkers	Baseline Median (IQR)	1-Month Median (IQR)	P^1^	P^2^	1-Year Median (IQR)	P^1^	P^2^
**Bone panel**							
AFOS (U/L)				0.554			**<0.001 ***
Placebo	64 (53, 76)	63 (58, 77)	0.480		61 (55, 82)	0.528	
ZA	71 (58, 80)	67 (62, 78)	0.826		57 (49, 68)	**<0.001**	
RANKL (pg/mL)				0.369			0.724
Placebo	11.3 (0.0, 49.2)	14.6 (0.0, 67.2)	0.470		11.3 (0.0, 59.6)	0.520	
ZA	13.0 (0.0, 37.0)	0.0 (0.0, 41.3)	0.891		14.4 (0.0, 48.5)	0.853	
iPINP (ng/mL)				**0.018**			**<0.001 ***
Placebo	35 (27, 40)	36 (23, 42)	0.729		38 (26, 45)	0.123	
ZA	36 (27, 49)	29 (20, 33)	**0.004**		15 (13, 18)	**<0.001**	
CTX-1 (pg/mL)				**0.009**			0.211
Placebo	0.2 (0.1, 0.4)	0.2 (0.1, 0.6)	0.546		0.3 (0.2, 0.4)	0.189	
ZA	0.3 (0.2, 0.5)	0.0 (0.0, 0.0)	0.154		0.2 (0.1, 5.4)	0.927	
**Chemokine panel**							
Eotaxin-1 (pg/mL)				0.841			0.398
Placebo	214 (136, 263)	202 (151, 258)	0.956		193 (149, 239)	0.622	
ZA	163 (146, 201)	167 (140, 224)	0.841		180 (144, 232)	0.546	
Eotaxin-3 (pg/mL)				0.565			0.862
Placebo	17.9 (10.5, 22.7)	16.0 (10.4, 19.6)	0.245		13.9 (11.9, 19.9)	0.261	
ZA	20.2 (12.2, 31.6)	18.3 (11.5, 27.6)	0.076		22.3 (12.9, 31.7)	0.388	
IP-10 (pg/mL)				0.056			**0.005 ***
Placebo	265 (199, 349)	240 (204, 351)	0.756		248 (171, 305)	0.177	
ZA	253 (187, 305)	282 (231, 423)	**0.036**		316 (240, 420)	**0.015**	
MIP-1A (pg/mL)				0.518			0.817
Placebo	13.5 (5.3, 21.7)	16.7 (5.3, 21.5)	0.818		16.7 (5.3, 19.1)	0.804	
ZA	5.3 (5.3, 16.3)	5.3 (5.3, 15.7)	0.945		5.3 (5.3, 17.5)	0.719	
MIP-1B (pg/mL)				0.883			0.091
Placebo	126 (79, 156)	102 (91, 161)	0.729		106 (83, 158)	0.177	
ZA	85 (58, 144)	87 (63, 136)	0.330		85 (66, 156)	0.277	
MCP-1 (pg/mL)				0.327			0.174
Placebo	276 (226, 337)	286 (249, 332)	0.812		267 (222, 346)	0.756	
ZA	237 (207, 281)	265 (207, 313)	**0.040**		279 (206, 326)	0.133	
MCP-4 (pg/mL)				0.925			0.429
Placebo	158 (108, 189)	140 (111, 190)	0.674		135 (109, 164)	0.701	
ZA	136 (105, 172)	133 (115, 180)	0.571		136 (108, 193)	0.330	
MDC-1 (ng/mL)				0.445			0.925
Placebo	1.1 (0.8, 1.4)	1.1 (0.8, 1.4)	0.498		1.1 (0.7, 1.4)	0.546	
ZA	1.1 (0.9, 1.3)	1.0 (0.9, 1.2)	0.571		1.0 (0.9, 1.2)	0.701	
RANTES (ng/mL)				0.529			0.968
Placebo	78 (48, 168)	85 (52, 116)	0.245		81 (62, 126)	0.261	
ZA	72 (50, 163)	80 (41, 126)	0.097		81 (54, 113)	0.648	
TARC (pg/mL)				0.157			0.512
Placebo	295 (197, 391)	326 (229, 438)	0.216		329 (198, 411)	0.231	
ZA	202 (179, 391)	241 (169, 344)	0.409		269 (161, 383)	0.812	
MIG-1 (pg/mL)				0.908			0.122
Placebo	72 (65, 118)	72 (37, 154)	0.880		72 (33, 118)	0.446	
ZA	69 (28, 132)	78 (33, 118)	0.939		69 (47, 134)	0.262	
**Cytokine panel**							
IL-7 (pg/mL)				0.738			0.445
Placebo	18.0 (14.0, 22.8)	15.8 (11.1, 20.9)	0.452		14.4 (11.4, 17.7)	0.133	
ZA	19.3 (15.2, 23.7)	16.2 (13.9, 20.1)	0.154		18.7 (13.9, 21.0)	0.064	
IL-12/23p40 (pg/mL)				0.341			0.758
Placebo	122 (92, 157)	113 (87, 185)	0.571		116 (94, 154)	0.898	
ZA	102 (77, 145)	97 (84, 134)	0.546		100 (84, 135)	>0.999	
IL-15 (pg/mL)				0.134			0.678
Placebo	2.8 (2.3, 3.4)	2.6 (2.4, 3.3)	0.985		2.8 (2.5, 3.2)	0.869	
ZA	3.0 (2.5, 3.2)	3.1 (2.6, 3.4)	0.123		3.0 (2.7, 3.3)	0.312	
IL-16 (pg/mL)				0.301			**0.035**
Placebo	191 (151, 246)	194 (167, 213)	0.515		172 (156, 200)	**0.027**	
ZA	173 (151, 223)	205 (174, 220)	0.349		207 (167, 248)	0.245	
IL-17A (pg/mL)				0.201			0.841
Placebo	2.3 (1.7, 3.3)	2.6 (1.5, 3.3)	0.177		2.6 (1.5, 3.5)	0.869	
ZA	2.2 (1.5, 3.3)	2.3 (1.6, 2.9)	0.622		2.2 (1.4, 3.4)	0.829	
TNF-B (pg/mL)				0.175			0.636
Placebo	0.4 (0.1, 0.5)	0.4 (0.1, 0.5)	0.761		0.3 (0.1, 0.5)	0.940	
ZA	0.1 (0.1, 0.4)	0.1 (0.1, 0.4)	0.240		0.3 (0.1, 0.4)	0.588	
**Anti-inflammatory panel**							
IL-1sRII (ng/mL)				0.989			0.391
Placebo	22.1 (18.8, 28.1)	23 (15, 26)	0.515		23 (16, 27)	0.651	
ZA	24.7 (20.5, 28.0)	22 (19, 28)	0.522		25 (21, 27)	0.181	
**Pro-inflammatory panel**							
IFN-A (pg/mL)				0.329			0.267
Placebo	71 (55, 91)	66 (47, 87)	0.806		66 (55, 95)	0.924	
ZA	61 (55, 78)	66 (55, 78)	0.070		71 (58, 79)	**0.032**	
IFN-G (pg/mL)				0.820			0.445
Placebo	5.6 (4.3, 9.3)	4.7 (3.2, 9.6)	0.498		4.6 (3.7, 8.0)	0.245	
ZA	4.5 (3.3, 7.1)	5.5 (3.2, 8.8)	0.898		5.3 (3.1, 8.3)	0.756	
IL-2R (pg/mL)				0.425			0.625
Placebo	144 (95, 229)	164 (109, 214)	0.225		123 (89, 193)	0.671	
ZA	117 (97, 163)	117 (90, 161)	0.588		125 (97, 150)	0.899	
IL-6 (pg/mL)				0.445			0.192
Placebo	0.8 (0.5, 1.1)	0.6 (0.5, 1.1)	0.648		0.7 (0.4, 0.8)	0.105	
ZA	0.7 (0.4, 0.9)	0.6 (0.5, 1.0)	0.674		0.6 (0.5, 0.9)	0.522	
IL-8 (pg/mL)				0.583			0.659
Placebo	11.1 (7.5, 14.9)	10.2 (7.9, 11.6)	**0.040**		9.2 (6.8, 12.2)	0.165	
ZA	9.7 (7.5, 11.7)	9.7 (7.4, 14.2)	0.294		10.0 (7.6, 13.7)	0.546	
TNF-A (pg/mL)				0.512			**0.049**
Placebo	2.0 (1.6, 2.3)	2.0 (1.7, 2.2)	0.231		1.8 (1.4, 2.2)	0.294	
ZA	1.6 (1.2, 2.2)	1.7 (1.3, 2.3)	0.956		1.7 (1.2, 2.5)	0.202	
**Angiogenesis panel**							
VEGF-A (pg/mL)				0.758			0.947
Placebo	249 (110, 388)	198 (125, 320)	0.596		182 (111, 406)	0.729	
ZA	380 (167, 530)	348 (178, 502)	0.189		412 (170, 497)	0.841	
VEGF-C (pg/mL)				0.947			0.369
Placebo	417 (342, 498)	459 (288, 510)	0.812		432 (330, 548)	0.245	
ZA	392 (382, 506)	399 (369, 501)	0.596		403 (346, 514)	0.869	
VEGF-D (ng/mL)				0.529			0.495
Placebo	0.7 (0.6, 1.0)	0.8 (0.6, 1.0)	0.349		0.8 (0.6, 0.9)	0.105	
ZA	0.7 (0.6, 0.9)	0.7 (0.6, 0.8)	0.596		0.7 (0.6, 0.9)	0.571	
Tie-2 (ng/mL)				0.529			0.314
Placebo	4.6 (4.0, 5.2)	4.7 (3.7, 5.5)	0.869		4.5 (3.9, 5.1)	0.701	
ZA	4.4 (4.0, 5.3)	4.4 (3.7, 5.1)	0.245		4.3 (3.5, 4.9)	**0.019**	
Flt-1 (pg/mL)				0.925			0.989
Placebo	88 (77, 97)	91 (63, 126)	0.812		90 (80, 99)	0.756	
ZA	91 (75, 98)	83 (76, 93)	0.349		82 (78, 99)	0.756	
bFGF (pg/mL)				0.659			0.659
Placebo	3.4 (1.9, 5.1)	3.2 (2.3, 5.3)	0.869		3.5 (1.9, 8.0)	0.123	
ZA	5.2 (2.7, 7.3)	3.7 (2.9, 7.1)	0.522		4.5 (3.8, 8.3)	0.522	
HGF-1 (pg/mL)				0.974			0.201
Placebo	205 (110, 263)	184 (115, 248)	0.874		157 (84, 248)	0.245	
ZA	146 (89, 219)	143 (67, 193)	0.374		171 (89, 248)	0.430	
**Vascular injury panel**							
hs-CRP (mg/L)				0.176			0.551
Placebo	1.2 (0.4, 2.6)	0.9 (0.6, 1.9)	0.936		0.9 (0.6, 2.0)	0.157	
ZA	0.9 (0.5, 1.2)	1.3 (0.4, 2.2)	**0.042**		0.8 (0.4, 1.3)	0.652	
ICAM-1 (µg/mL)				0.341			0.773
Placebo	0.4 (0.4, 0.5)	0.4 (0.3, 0.5)	0.368		0.4 (0.3, 0.5)	0.196	
ZA	0.4 (0.3, 0.5)	0.4 (0.3, 0.5)	0.622		0.3 (0.3, 0.4)	0.073	
VCAM-1 (µg/mL)				0.547			0.603
Placebo	0.6 (0.6, 0.8)	0.6 (0.5, 0.7)	0.409		0.6 (0.5, 0.7)	0.096	
ZA	0.6 (0.5, 0.7)	0.5 (0.5, 0.7)	0.841		0.5 (0.5, 0.7)	0.369	
SAA (µg/mL)				0.512			0.954
Placebo	1.8 (1.1, 4.1)	1.9 (1.2, 4.2)	0.452		1.7 (1.3, 2.5)	0.595	
ZA	1.7 (0.8, 2.2)	1.9 (1.0, 4.3)	**0.048**		1.5 (0.6, 2.8)	0.922	

^1^ Significance in Wilcoxon signed rank test, ^2^ Significance in Mann–Whitney’s test; * significant after B–H correction; AFOS = alkaline phosphatase; bFGF = basic fibroblast growth factor beta; CTX1 = C telopeptide of type I collagen; Flt = fms related tyrosine kinase (=VEGFR1); HGF = hepatocyte growth factor; hs-CRP = high-sensitive C-reactive protein; ICAM = intercellular adhesion molecule; IL = interleukin; IFN = interferon; IP = interferon-γ-inducible protein; iPINP = intact procollagen I N-terminal propeptide; MCP = monocyte chemotactic protein; MDC = macrophage derived chemokine; MIG = monokine induced by gamma-interferon; MIP = macrophage inflammatory protein; RANKL = receptor activator of NF-κB ligand; RANTES = regulated upon activation, normally T-expressed, and presumably secreted; SAA = serum amyloid A; TARC = thymus and activation-regulated chemokine; Tie = TEK receptor tyrosine kinase; TNF = tumor necrosis factor; VCAM = vascular cell adhesion molecule; VEGF = vascular endothelial growth factor.

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
