# Peer review of "The Effect of Zoledronic Acid on Serum Biomarkers among Patients with Chronic Low Back Pain and Modic Changes in Lumbar Magnetic Resonance Imaging"

_diagnostics, 2019, doi:10.3390/diagnostics9040212_

Round 1

Reviewer 1 Report

The goal of the study was to compare changes in serum biomarkers including 33 inflammatory mediators, signaling molecules, growth factors and markers of bone turnover after 34 a single intravenous infusion of 5 mg zoledronic acid (ZA, a long-acting bisphosphonate; n=20) or 35 placebo (n=20) among patients with Modic change (MC) and chronic low back pain in a randomized 36 controlled design. After 1 year they observed significant differences in
39 three biomarkers over one year; interferon--inducible protein (IP-10) had risen in the ZA group 40 (p=0.005), whereas alkaline phosphatase (AFOS) and intact procollagen I N-terminal propeptide 41 (iPINP) had significantly decreased in the ZA group.

Author Response

REPLY TO REVIEWERS

Diagnostics-631283- The effect of zoledronic acid on serum biomarkers among patients with chronic low back pain and Modic changes in lumbar magnetic resonance imaging

Dear Editor,

The author would like to thank you and the reviewers for their time and effort devoted to the review of our manuscript. The comments were insightful and greatly appreciated. As such, the authors would like to take this opportunity to address each concern the Reviewers noted in their review of our submission. In addition, where appropriate, we have revised our manuscript accordingly.

The additions to the manuscript are highlighted in yellow. Unfortunately, the deletions to the text are not shown in order to improve the readability of the manuscript.

We have rewritten some sentences which were identical with expressions used in our former articles.

We believe that the Reviewers’ comments have improved the quality of our manuscript. We hope that you and the Reviewers will now find our manuscript suitable for publication in the Diagnostics.

The responses to the comments of Reviewer II

Specific comments:

Comment 1: The MRI spectra from pre-infusion (baseline imaging experiments) and post infusion as well as from follow-up experiment would improve the quality of the work. 

Response: The field strength and fat saturation modify the T1 and T2 relaxation times and tissue visualization. All these effects influence on the appearance of water and fat signals, which distinguish type 1 Modic changes (MC1) from type 2 lesions (MC2) and are integral parts of MC1 and MC2. A Danish study reported that a higher number of MC was diagnosed more likely in a high-field scanner and the distribution of MC1 and MC2 varied between high- and low-field scanners; MC1 were 3-4-times more prevalent in 0.3T MRI scanners and MC2 were detected twice as often with 1.5T MRI scanners (Bendix et al. Lumbar modic changes -a comparison between findings at low- and high-field magnetic resonance imaging. Spine. 2012;37(20):1756-62). Recently, imaging and reporting guidelines for MC were published (Fields et al. Measuring and reporting of vertebral endplate bone marrow lesions as seen on MRI (Modic changes): recommendations from the ISSLS Degenerative Spinal Phenotypes Group. Eur Spine J. 2019;28:2266-74 ). We included the description of MRI instruments and magnetic field strengths in the manuscript.

Comment 2:

It would also be useful to have a figure showing the changes in serum biomarker pre- and post- Zoledronic acid treatment. 

Response: We have added figures as supplementary material.

Reviewer 2 Report

This manuscript reports a study for comparing changes in such as signaling molecules, inflammatory mediators and growth factors following a single  intravenous dose of Zoledronic acid or placebo in patients suffering from Modic change (MC) and chronic low back pain. The authors  measured the 39 serum biomarkers, pre and post (and month and one year) Zoledronic acid treatment and observed significant changes in interferon-γ-inducible protein, alkaline phosphatase and intact procollagen I N-terminal propeptide. The proposed study is very relevant as it provides a positive correlation with Zoledronic acid and Modic change and how it is reflected in serum which can further lead to improved diagnostic and therapeutic approaches as well as treatment monitoring. 

I am very impressed with the way the study was designed,experiments conducted and the MRI as well as the serum biomarker analysis was performed to investigate this challenging research problem.

The manuscript is well written and I strongly recommend publication of this manuscript with some minor revisions. Suggestions for revision are included.

The MRI spectra from pre-infusion (baseline imaging experiments) and post infusion as well as from follow-up experiment would improve the quality of the work. 

It would also be useful to have a figure showing the changes in serum biomarker pre- and post- Zoledronic acid treatment. 

Author Response

(The authors gave the same response as above.)
